# Mapping of QTLs Associated with Biological Nitrogen Fixation Traits in Peanuts (*Arachis hypogaea* L.) Using an Interspecific Population Derived from the Cross between the Cultivated Species and Its Wild Ancestors

**DOI:** 10.3390/genes14040797

**Published:** 2023-03-26

**Authors:** Darius T. Nzepang, Djamel Gully, Joël R. Nguepjop, Arlette Zaiya Zazou, Hodo-Abalo Tossim, Aissatou Sambou, Jean-François Rami, Valerie Hocher, Saliou Fall, Sergio Svistoonoff, Daniel Fonceka

**Affiliations:** 1Centre d’Etudes Régional pour l’Amélioration de l’Adaptation à la Sécheresse, CERAAS-Route de Khombole, Thiès BP 3320, Senegal; 2PHIM Plant Health Institute, Univ Montpellier, IRD, CIRAD, INRAE, Institut Agro, Montpellier, France; 3Laboratoire Commun de Microbiologie (LCM) (IRD/ISRA/UCAD), Centre de Recherche de Bel Air, Dakar BP 1386, Senegal; 4Laboratoire Mixte International Adaptation des Plantes et Microorganismes Associés aux Stress Environnementaux (LAPSE), Centre de Recherche de Bel Air, Dakar CP 18524, Senegal; 5Dispositif de Recherche et de Formation en Partenariat, Innovation et Amélioration Variétale en Afrique de l’Ouest (IAVAO), CERAAS Route de Khombole, Thiès BP 3320, Senegal; 6CIRAD, UMR AGAP, F-34398 Montpellier, France; 7AGAP, Univ Montpellier, CIRAD, INRAE, Institut Agro, Montpellier, France; 8Institute of Agricultural Research for Development (IRAD) (IRAD), Maroua, Cameroon

**Keywords:** peanut, symbiosis, wild species, sustainable agriculture, plant nutrition

## Abstract

Peanuts (*Arachis hypogaea* L.) are an allotetraploid grain legume mainly cultivated by poor farmers in Africa, in degraded soil and with low input systems. Further understanding nodulation genetic mechanisms could be a relevant option to facilitate the improvement of yield and lift up soil without synthetic fertilizers. We used a subset of 83 chromosome segment substitution lines (CSSLs) derived from the cross between a wild synthetic tetraploid AiAd (*Arachis ipaensis* × *Arachis duranensis*)^4×^ and the cultivated variety Fleur11, and evaluated them for traits related to BNF under shade-house conditions. Three treatments were tested: without nitrogen; with nitrogen; and without nitrogen, but with added0 *Bradyrhizobium vignae* strain ISRA400. The leaf chlorophyll content and total biomass were used as surrogate traits for BNF. We found significant variations for both traits specially linked to BNF, and four QTLs (quantitative trait loci) were consistently mapped. At all QTLs, the wild alleles decreased the value of the trait, indicating a negative effect on BNF. A detailed characterization of the lines carrying those QTLs in controlled conditions showed that the QTLs affected the nitrogen fixation efficiency, nodule colonization, and development. Our results provide new insights into peanut nodulation mechanisms and could be used to target BNF traits in peanut breeding programs.

## 1. Introduction

Nitrogen is one of the most limiting nutrients in agriculture. Species of the legume family can associate with beneficial soil bacteria called “rhizobia”, which reduce atmospheric nitrogen (N_2_) into nitrogen assimilable by the plant in exchange for a supply of carbon [1]. A legume–rhizobia symbiosis results in biological nitrogen fixation (BNF) inside a new organ called a nodule. This symbiosis provides approximately 40 million tons of nitrogen (N) annually to agricultural systems [2]. In farming systems using low amounts of synthetic fertilizers derived from fossil fuels (Haber–Bosh), BNF is key to the growth and production of legumes and also serves as a significant N source for associated or subsequent crops such as cereals [3]. Deciphering the genetic mechanisms that regulate this biological process is essential for its optimization and for the development of sustainable agriculture. However, although BNF is a well-studied process in model legumes and a few crops, breeding programs have rarely targeted this trait.

Domestication is the selection, modification, and adoption of wild plant species with traits of interest to humans [4]. Over 12,000 years ago, hundreds wild plant species have been transformed into improved crop plants [5]. Following domestication, modern agriculture practices have allowed yields to increase considerably, but have also led to a high reduction in the genetic diversity of cultivated plants as compared to wild relatives of crops [6,7]. A major consequence of the recent human selection pressure on crops is the alteration of the beneficial plant–microorganism associations in the soil and, consequently, an increase in the dependency on synthetic fertilizers, which are costly for farmers and harmful to the environment [8]. Mutch and Young [9] showed in peas (*Pisum sativum*) and broad beans (*Vicia faba* L.) that cultivated species are less promiscuous in their interaction with indigenous rhizobia compared to wild species. A reduction in host range was also reported between cultivated and wild species for chickpeas (*Cicer arietinum* L.) [10] and soybeans (*Glycine max* L.) [11], suggesting that the domestication process and human selection have affected the compatibility of the host–rhizobia interaction. Kiers et al. [12] showed that old soybean cultivars produce more seeds than newer cultivars when infected with a mixture of effective and ineffective rhizobia, suggesting that the older cultivars have an ability to select best-matching partners from a pool of compatible rhizobia. In peas, alfalfa (*Medicago sativa* L.), and common beans (*Phaseolus vulgaris* L.), the wild species and landraces exceed the improved varieties in nitrogen fixation efficiency [13]. These results suggest that wild legume species could be used as reservoirs of favorable alleles to improve the symbiotic potential of their cultivated counterparts [14,15]. In contrast, Muñoz et al. [16] reported that cultivated soybeans performed better than wild soybeans regarding BNF and identified quantitative trait loci (QTLs) for traits related to BNF, which showed a very low diversity among cultivated soybeans, possibly as a result of human selection during the domestication of soybeans. The overall impact of domestication on the symbioses of legume crops is difficult to assess, as very little information is available for other species.

Cultivated peanuts (*A. hypogaea* L.) are an oilseed legume crop that can meet up to 68% of its nitrogen needs via BNF [2], thanks to its interaction with rhizobia of the genus *Bradyrhizobium* [17,18]. Despite the low genetic diversity at the DNA level [19], several studies have reported phenotypic variations within the cultivated compartment of peanuts for key agronomic traits, including the ones linked to BNF such as the nodule weight and number, the biomass, nitrogenase activity, the total nitrogen content, and the amount of fixed nitrogen [20,21,22,23]. These studies suggest that the symbiotic potential of peanuts could be enhanced by exploiting the genetic variation in these traits. However, as BNF happens in nodules that are underground organs, it is not a straightforward trait to phenotype for; consequently, it remains difficult to target in breeding programs. In addition, BNF is subjected to environmental variation, meaning that multiple years and location trials are needed to better capture the trait variation. These constraints contribute to a limited knowledge of the genetic mechanisms linked to the nodulation of peanuts and other legume crops. The genetic and morphological characterization of non-nodulating and nodulating lines in peanuts suggests that the nodulation process is controlled by several genes [24,25,26]. Microsynteny and a transcriptome analysis showed that orthologs of almost all the symbiotic genes described in model legumes are present in *Arachis* sp. [27,28,29,30]. Peanuts belong to the dalbergioid clade, where rhizobia invade plant roots intercellularly through natural fissures at the lateral root base called “cracks” [31,32,33]. The molecular basis of this mode of rhizobia infection remains poorly understood.

Peanuts appeared about 9400 years ago from an interspecific hybridization between two wild diploid species, *A. duranensis* (genome A) and *A. ipaensis* (genome B), followed by a duplication of the chromosomes [34]. The combined effects of this single tetraploidization event, domestication, and selection have greatly narrowed its genetic basis. Advances in genetics and genomics offer opportunities to dive into feral peanut diversity to mine new alleles for breeding [35,36]. Interspecific mapping populations such as advanced backcross (AB-QTL) and chromosome segment substitution lines (CSSLs) were developed and used in peanuts to identify several QTLs related to yield components, disease resistance, oil quality, and plant morphology [37,38,39,40]. However, to our knowledge, interspecific mapping populations have not been exploited to investigate the effect of wild species alleles on BNF.

In this study, we used a CSSL population derived from the cross between a cultivated peanut variety and a synthetic tetraploid, created by gathering the genomes of the most probable wild diploid ancestors of the cultivated species, to target traits related to BNF. Genotype–phenotype associations allowed for the identification of QTLs for BNF variation. Our results provide new insights into the genetics of peanut nodulation and useful information for pre-breeding programs.

## 2. Materials and Methods

### 2.1. Plant Materials

A subset of 83 CSSLs was selected from a population of 122 CSSLs (BC_4_F_3_) developed at CERAAS (Centre d’Etudes Régional de Recherche pour l’Amélioration de l’Adaptation à la Sécheresse). These CSSLs were derived from an interspecific cross between a cultivated Fleur11 variety, used as the female parent, and the synthetic tetraploid AiAd, which is a combination of the genomes of (*A. ipaensis* KG30076 (BB genome) × *A. duranensis* V14167 (AA genome))^4×^, used as the male parent [38]. Fleur11 is a Spanish-type variety with a short cycle (90 days) that is cultivated in West Africa for its agronomic performance and its moderate tolerance to drought. The selected CSSLs constitute the minimum set of lines that provide optimal genomic coverage. The wild introgression fragments have an average length of ~39.2 cM and cover ~88.7% of the genetic map. Information regarding the genotyping and markers used to develop this population is available in our previous work [38].

### 2.2. Phenotyping of CSSL Population for Traits Associated with BNF

The 83 CSSLs and the recurrent parent Fleur11, used as a control, were evaluated for their BNF capacity under shade-house conditions in December 2017 and March 2018 in Senegal. Seeds were sterilized with calcium hypochlorite and germinated in Petri dishes, and seedlings were transferred manually into pots containing a leached and sterilized substrate with a low nitrogen content, as described in [18]. Three treatments were applied: (i) without nitrogen (negative control or −N), where no external nitrogen supply was added to the substrate; (ii) with nitrogen (positive control or +N), where the equivalent of 100 kg/ha of urea was applied in two stages (at one and six weeks) to each pot; and (iii) without nitrogen + inoculation, where 5 mL of an inoculum of *B. vignae* strain ISRA400 (−N + ISRA400) was added to each pot. ISRA400 is an efficient bacterial strain that was isolated from the nodules of peanut plants grown in Senegal [18]. The inoculum was produced by cultivating the bacteria in yeast extract mannitol medium [41] in the dark for 4 days at 28 °C, shaken at 250 rpm, until obtaining cell suspensions with an absorbance of ~0.7 at 600 nm. The inoculum was applied one week after the seedlings were transferred to the pots. The experimental design used was an α-lattice with 3 replications and 6 blocks of 15 genotypes per replication. The studied factor was the “genotype” with 90 levels (83 CSSLs and the recurrent parent Fleur11 repeated 7 times), totaling 270 plants per treatment.

For each treatment, the leaf chlorophyll content and total dry biomass were evaluated. The leaf chlorophyll content was indirectly estimated weekly on the third youngest leaf using SPAD (soil plant analysis development; Konica-Minolta, Okasaka, Japan). After nine weeks, the plants were harvested and dried at 65 °C for 72 h before weighing to obtain the total dry biomass.

### 2.3. Statistical Analysis and QTL Mapping

Descriptive statistics (mean and standard deviation) were calculated for each trait and each treatment in both experiments. The normality and homogeneity of variance were checked with the Kolmogorov–Smirnov and Levene tests, respectively. An analysis of variance (ANOVA) was performed for each experiment to estimate the genotypes and replication effects on each trait using a standard linear model:*y_ijk_* = *µ* + *G_i_* + *r_j_* + *b_jk_* + *ɛ_ijk_*
where *y_ijk_* = observed value of genotype *i* in replication *j* and block *k*; *µ* = mean value of the population; *G_i_* = the effect of genotype *i*; *r_j_ =* the effect of replication *j*; *b_jk_* = the effect of block *k* in replication *j*; and *ɛ_ijk_* = residual error.

Estimates of broad-sense heritability were calculated for each treatment in each environment using the formula:h^2^ = 1 − (1/F) 
where F is the F-value associated with the genotype effect [42].

When a significant genotype effect was found for a given trait, each CSSL was compared to the recurrent parent Fleur11 with Dunnett’s multiple comparisons test at *p* < 0.05 using the package multcomp of the R software [43], as described in [38]. When a given CSSL carried one unique wild chromosome segment (i.e., target wild segment) and showed a significant difference compared to Fleur11, we assumed that the difference was linked to the presence of the wild segment, and consequently, that at least one QTL was located in the wild segment. When a given line harbored the target wild segment and other segments elsewhere in the genome, the QTL assignment was performed in two steps. We first compared the phenotype of the line with neighboring lines that had overlapping fragments. If the phenotypes were similar, the QTL was attributed to the target wild chromosome segment. If the phenotypes were different, the comparison was made with lines that carry similar regions elsewhere in the genome. In some particular cases, the genotype/phenotype comparison did not allow for the unambiguous assignment of a QTL in one wild fragment; therefore, the QTL was considered a putative QTL and was associated with all the wild chromosome fragments present in that line.

The QTL effect for a given trait was calculated according to the formula below [38]:$$QTL\;effect\left(\%\right)=\frac{\left[Mean\left(CSSL\right)-Mean\left(Fleur11\right)\right]\times 100}{Mean\left(Fleur11\right)}$$

### 2.4. Cytological Analysis of Nodules

A cytological analysis was performed to provide information about the QTL effect on the shape, structure, and functioning of nodules. Fresh nodules obtained in controlled conditions were harvested, embedded in 4% agar, and sliced into 80 µm sections using a VT1000S vibratome (Leica Biosystems, Nussloch, Germany). The sections were successively incubated in live/dead staining solutions (5 μM SYTO 9 and 30 μM propidium iodide in 50 mM Tris (pH 7.0) buffer; Live/Dead BacLight, Invit-rogen, Carlsbad, CA, USA) for 15 min, and calcofluor white M2R (0.01% wt/vol in 10 mM phosphate saline buffer, (Sigma, Germany) for 15 min in the dark [44]. The stained sections were mounted in saline–glycerol phosphate buffer (*vol*/*vol*) and observed using an LSM 700 confocal laser scanning microscope (Carl Zeiss, Germany). SYTO9 and propidium iodide were excited to 488 and 555 nm for an emission range of 490–522 nm and 555–700 nm, respectively. For calcofluor, the excitation wavelength was 405 nm and the emission range was 405–470 nm. Nodule sections were also observed under an AZ differential interference microscope (Nikon, Tokyo Japan) equipped with a DS-Ri2 camera to describe the nodule structure.

## 3. Results

### 3.1. Phenotypic Variation in BNF in the CSSL Population under Shade-House Conditions

In order to investigate the phenotypic variations in BNF, the CSSL population was grown in a shade house under three treatments (−N: without nitrogen, +N: with nitrogen, and −N+ISRA400: without nitrogen + *B*. *vignae* ISRA400), and two traits related to BNF, the leaf chlorophyll content and the biomass, were recorded. The experiment was conducted in two consecutive environments (December 2017 and March 2018). We observed a gradual decrease in the average leaf chlorophyll content in the negative control treatment (−N), while the chlorophyll content remained almost constant in the positive control treatment (+N) (Appendix A). In the inoculated plants (−N+ISRA400), the leaf chlorophyll content decreased during the three first weeks, and then started to increase 4 weeks after inoculation, indicating the establishment of a functional symbiotic interaction that supplied nitrogen to the plants (Appendix A). The fertilized and inoculated plants produced more biomass in comparison to the −N treatment (Table 1). The results of the ANOVAs showed a significant genotype effect for the chlorophyll content only in the +N and −N+ISRA400 treatments, while for the total biomass, a significant genotype effect was observed in all treatments (Table 1). The broad-sense heritability was estimated for each trait in each environment. We observed null, moderate, and high broad-sense heritability estimates for the chlorophyll content in the −N, +N, and −N+ISRA400 treatments, respectively, in both environments. For the total biomass, a moderate heritability was found for all the treatments in both environments (Table 1).

### 3.2. QTL Mapping

To search for significant line × trait associations in each environment, the CSSLs were compared to Fleur11. We considered a significant line × trait association to be specifically linked to biological nitrogen fixation when it was not observed in the control treatments.

#### 3.2.1. Chlorophyll Content

In the 2017 environment, the Dunnett’s test showed that six lines had a significantly lower leaf chlorophyll content compared to Fleur11 in the inoculated (−N+ISRA400) treatment, while in the negative (−N) and positive (+N) control treatments, no significant difference was observed (Appendix A). The relative differences between these lines and Fleur11 ranged from −24.61 to −65.29% (Appendix A). When the experiment was repeated in 2018, a total of eight lines had a significantly lower leaf chlorophyll content compared to Fleur11 in the inoculated treatment (Appendix A). The relative differences between these lines and Fleur11 ranged from −15.17 to −73.49% (Appendix A). One line (12CS_010) had a significantly higher chlorophyll content compared to Fleur11 in the +N treatment. All associations that were significant only under inoculated conditions were summarized and used to identify the QTLs in each environment. In the 2017 environment, eight QTLs were linked to the leaf chlorophyll content (Table 2). One QTL was unambiguously located on each of the linkage groups (LGs) A02, A03, A04, A08, and B02 (Figure 1). In some particular cases, we were not able to unambiguously associate a QTL to one unique wild chromosome segment; therefore, the QTL was considered a putative QTL and was associated with all the fragments present in that line. This was, for example, the case for line 12CS_060, which had overlapping wild segments with several lines on the LGs A07, B06, and B08 (Figure 1). In 2018, eleven genomic regions distributed on nine LGs were identified (Table 2). One QTL was clearly mapped on each of the LGs A02, A03, A07, A08, and B02 (Figure 1). Six QTLs on the LGs A03, A04, A08, A09, B01, and B10 were considered putative QTLs. Finally, four QTLs located on the LGs A02, A03, A08, and B02 were consistently mapped in both environments.

#### 3.2.2. Total Biomass (TB)

The comparison of the total biomass between the CSSLs and Fleur11 in 2017 showed that six lines had a significantly lower biomass than Fleur11 in the −N treatment, and 26 lines had a significantly lower biomass in the inoculated treatment (Appendix A). However, among the lines found in the inoculated treatment, three lines (12CS_004, 12CS_042, and 12CS_098) were also found in the negative control treatment. Finally, a total of 23 lines were identified only in the inoculated treatment, with the relative differences ranging from −39.73 to −71.51% (Appendix A). In 2018, significant differences were also found between several lines and Fleur11: twenty-three lines in the −N treatment, one line in the +N treatment, and nine lines in the inoculated treatment (Appendix A). Among the nine lines detected in the inoculated treatment, four (12CS_004, 12CS_034, 12CS_118, and 12CS_081) were also identified in the −N treatment. Finally, five lines were identified only in the inoculated treatment, with relative differences ranging from −55.10 to −60.58% (Appendix A). Regarding the mapping of QTLs in the 2017 environment, the line × trait associations allowed the identification of 20 QTLs located on fifteen linkage groups (Table 2). Three QTLs were mapped on LG A01, two QTLs on LG B07, and one QTL on each of the LGs A02, A05, A08, B02, B02, B04, B06, B08, B10, and B11 (Figure 1). Six QTLs were considered putative QTLs: two QTLs on LG A09 and one QTL on each of the LGs A04, A06, B03, and B10. In the 2018 environment, six QTLs located on five LGs were identified (Table 2). One QTL was unambiguously mapped on each of the LGs A02, A08, and B02 (Figure 1), whereas three QTLs located on the LGs A03, A04, and A08 were considered putative QTLs. Finally, three QTLs on the LGs A02, A08, and B02 were consistently identified in both environments.

In summary, 15 QTLs were identified for the leaf chlorophyll content and 25 QTLs were identified for the total biomass in 2017 and 2018 (Appendix A). Among these QTLs, four QTLs for the leaf chlorophyll content were stable (i.e., mapped in 2017 and 2018) and unambiguously mapped on the LGs A02, A03, A08, and B02, and three QTLs for the total biomass were stable and unambiguously mapped on the LGs A02, A08, and B02.

### 3.3. Validation of QTLs in Controlled Conditions

Five CSSLs that showed contrasting phenotypes compared to Fleur11 were selected to validate the QTLs. Among these lines, four (12CS_004, 12CS_044, 12CS_051, and 12CS_084) contained less chlorophyll and produced less biomass in the inoculated treatment, and one line (12CS_114) showed higher values for these traits, although these differences were not statistically significant (Appendix A).

#### Detailed Analysis of the Response to Inoculation

Differences in the plant growth between Fleur11 and the CSSLs were observed thirty-five days post-inoculation. For the leaf chlorophyll content, no difference was found between Fleur11 and the CSSLs under the −N treatment, while in the inoculated treatment, we observed a significant decrease in the chlorophyll content (~53–75%) for the lines 12CS_004, 12CS_044, 12CS_051, and 12CS_084 in the first and second experiments (Figure 2).

Regarding the total biomass, except for line 12CS_004 in the first experiment, no difference was found between Fleur11 and the CSSLs under the −N treatment (Figure 3). In the inoculated treatment, we found that 12CS_114 produced more biomass compared to Fleur11, but this difference was statistically significant in only one experiment. Three lines (12CS_004, 12CS_044, and 12CS_051) exhibited a significantly lower biomass (23–43%) than Fleur11 in both experiments (Figure 3). Although the biomass decrease was less pronounced under controlled conditions, these results globally confirmed the observations made under the shade house, providing additional evidence that the corresponding QTLs are possibly involved in the symbiotic interaction with *B. vignae* ISRA400.

To further investigate the phenotypes of these five lines, we evaluated the traits directly related to nodulation as indicators of symbiotic performance: the nitrogenase activity using an acetylene reduction assay, the number of nodules, the nodule dry weight, and the ratio between the nodule dry weight and the nodule number (Figure 4). The correlation matrix showed positive and significant Pearson coefficients between the nodulation traits, the leaf chlorophyll content, and the total biomass (Table 3), suggesting a strong relation between nodule development and general plant growth.

Two independent experiments performed under controlled conditions showed that the line 12CS_004 exhibited a significant decrease in all the nodulation traits compared to Fleur11 (Figure 4). Similar results were also observed for the line 12CS_044, except for the nodule number. The line 12CS_084 produced as many nodules as Fleur11, but with a significant decrease in the nodule dry weight and nitrogenase activity. The line 12CS_051 showed a significantly lower nitrogenase activity compared to Fleur11 in both experiments. Moreover, in one experiment, we found a significant decrease in the nodule dry weight for line 12CS_051, and in the ratio between the nodule dry weight and the nodule number for line 12CS_084. No significant difference was found between Fleur11 and line 12CS_114 for all the nodulation traits.

During legume–rhizobia interactions, the symbiotic efficiency may depend on the nodule colonization by the rhizobia, and also on the development and functioning of the nodule. To better investigate the effect of QTLs, we examined histological sections of nodules at 35 days post-inoculation. In Fleur11, we observed spherical and determinate nodules of the aeschynomenoid type that exhibited central infected tissue corresponding to the fixation zone, surrounded by uninfected cells corresponding to the inner cortex or nodule parenchyma, the vascular bundles, and the outer cortex (Figure 5a). Similar anatomical features were observed for all CSSLs, including lines 12CS_004 and 12CS_044. However, the nodules of those lines were smaller compared to the ones formed by Fleur11 (Figure 5b–f). To further investigate the functional characteristics of these nodules, the bacteria were stained with a live/dead staining protocol and examined using confocal microscopy. For Fleur11, we observed numerous infected cells in the fixation zone. Those cells contained densely packed green-labelled spherical bacteroids, a landmark of functional nodules (Figure 5g,m) as reported by Sinharoy and DasGupta [45]. Similar features were observed for line 12CS_114, which was able to fix high amounts of nitrogen (Figure 5l,r). Lines with lower nitrogen fixation levels also exhibited numerous infected cells containing spherical bacteroids, but these were generally less dense and showed a higher proportion of red-labelled dead bacteroids compared to Fleur11 (Figure 5j,o–q). In line 12CS_051, we were able to detect zones where the nodule tissue collapsed, corresponding to a spontaneous degeneration of nodule tissue associated with defense responses as described by Gully et al. [46] and Songwattana et al. [47]. These observations indicate that the wild alleles at the QTLs can reduce the nodule number and/or size and also affect the maintenance of functional bacteroids in infected cells situated in the central tissue.

We screened the sequences of genomic regions corresponding to the QTLs using PeanutBase “https://peanutbase.org/gbrowse_peanut1.0 (accessed on 30 January 2023)” and Orthofinder “https://davidemms.github.io/ (accessed on 23 April 2021), and found 17 orthologs of the nodulation genes previously described in model legumes (Appendix A).

## 4. Discussion

Biological nitrogen fixation (BNF) is a valuable component of sustainable agriculture. Understanding the genetics of the interaction between plants and rhizobia can facilitate the enhancement of BNF in grain legumes. In this study, 83 chromosome segment substitution lines (CSSLs) were used to map the QTLs of the leaf chlorophyll content and the total biomass under three treatments (−N, +N, and −N+ISRA400), and to explore the contribution of wild species to the improvement of BNF in cultivated peanuts.

### 4.1. Variability in Traits Related to BNF and QTL Mapping

Genetic and phenotypic variations were found for the chlorophyll content and/or the total biomass in the CSSL population (Table 1). We hypothesize that these variations could be linked to the variability in nitrogen use efficiency (available or residual N uptake and transport) in the control treatments (−N and +N), and the nitrogen fixation efficiency in the inoculated treatment (−N + *B. vignae* ISRA400). Sahrawat et al. [48] showed that a non-nodulating peanut line is less efficient at acquiring N in comparison to nodulating genotypes, suggesting a relationship between nodulation and N-uptake abilities. Regarding the phenotyping for BNF, the leaf chlorophyll content has been successfully used as a surrogate trait to evaluate BNF in grain legumes, including peanuts [23,26,49,50,51].

We observed that some lines, such as 12CS_055 and 12CS_106, presented a total biomass that was significantly lower compared to Fleur11 in the −N treatment; however, in the treatment −N+ISRA400, these lines exhibited a total biomass similar to Fleur11 (Appendix A). These findings could be explained by a strong sensitivity to nitrogen deficiency or by a positive effect of the wild fragments on the symbiotic performance. Thereby, for QTL detection, we assumed that significant associations were specifically linked to BNF when the phenotypic variation(s) they induced were not observed in the control treatments. Dunnett’s test allowed for the identification of a total of twenty-four and nine lines that were significantly different from Fleur11 for the −N+ISRA400 treatment in the first and second shade-house experiments, respectively (Table 2). The variability in the response to inoculation observed between the two experiments was possibly due to temperature or light intensity differences in the shade-house trials (warmer days in 2017 compared to 2018). Several environmental factors, including the temperature, N concentration in the root zone, soil water content, and plant nutrient status (C and N), can affect biological nitrogen fixation in legumes [52]. Despite the variation in the experimental conditions between the 2017 and 2018 environments, four lines (12CS_051, 12CS_004, 12CS_084, and 12CS_044) had similar features for the chlorophyll content and/or the total biomass (Table 2).

In the 2017 and 2018 environments, the genotype × phenotype associations allowed for the identification of 15 and 25 QTLs for the leaf chlorophyll content and the total biomass, respectively (Appendix A). The QTLs were consistently mapped on the LGs A02, A03, A08, and B02 for the leaf chlorophyll content and/or the total biomass (Figure 1, Table 2). A co-localization for the two traits was found on the LGs A02, A08, and B02, in accordance with previous studies showing that a N deficiency reduces leaf photosynthesis and plant biomass accumulation in legumes [53] and other plants [54]. As with other economically and agronomically important traits, BNF is a complex trait regulated by several genes with different effects. Numerous QTLs involved in biological nitrogen fixation are described in other grain legumes, such as common beans [50,55], peas [56], and soybeans [16,57,58,59,60]. The simple-sequence repeat marker genotyping of two pairs of recombinant inbred lines and their parents allowed for the identification of 20 chromosome regions associated with the loss of the nodulation ability in peanuts [26]. Recently, map-based cloning and QTL-seq approaches using F_2_ peanut populations have allowed for the identification of syntenic regions, including a pair of homoeologous genes coding for GRAS transcription factors involved in nodulation and nodulation signaling pathway 2 (*NSP2*) in the chromosomes A08 and B07 [61]. Due to the differences in the population and marker types used, it is difficult to precisely match our results with those of previous genetic mapping experiments on peanut nodulation. Interestingly, we found that one QTL on chromosome A03 was linked to the total biomass in the inoculated treatment and in the negative control. This QTL is carried by line 12CS_004 (Appendix A). Our findings suggest that two independent genes located in the same region are involved, for instance, in soil N uptake/transport and BNF, respectively, or that a single gene is involved in either downstream events such as nitrogen transport, metabolism, or remobilization or in low-nitrogen sensing. Examples of such genes have been reported in the literature. It is, for example, the case for the nitrate reductase gene, which plays a role in both soil N uptake [62] and in early nodulation steps [63]. The interactions among N fixation, nitrate reduction, and ammonium assimilation in legumes are well reviewed [64].

### 4.2. QTL Action on Nodulation Traits

The experiments under controlled conditions allowed for the confirmation of the shade-house results (Appendix A) and pointed out a positive correlation between the surrogate traits, chlorophyll content and biomass, and the traits directly linked to nodulation (Table 3). We found that plants harboring wild alleles at the QTLs showed a reduced nodule number, a reduced nodule dry weight, and/or an alteration of nodule functioning, but all nodules, even small ones, exhibited similar anatomical features as those observed in the cultivated variety Fleur11. Moreover, all the genotypes exhibited determinate nodules of the aeschynomenoid type, as previously described in peanuts [65,66,67] and *Aeschynomene indica* [68]. Interestingly, we showed that these nodules contained numerous infected cells where we detected circular-shaped, well-differentiated bacteroids, indicating that the processes leading to bacterial recognition, nodule organogenesis, intracellular infection, and cell differentiation into bacteroids are functional. However, bacteroids were generally less dense in plant cells and often labelled in red, indicating the presence of dead bacteroids. It has been reported that in the legume–rhizobia symbiosis, the plant can sanction the rhizobia that fail to fix nitrogen inside the nodules by reducing the resource allocation (carbon), leading to decreased rhizobial fitness [69,70,71]. Furthermore, in line 12CS_051, we observed regions with collapsed necrotic tissue, which is often associated with incompatibility and can be assimilated to early nodule senescence triggered by plant defense mechanisms that target the invading bacteria. Necrotic zones and the presence of dead bacteria are associated with symbiotic incompatibilities and are characterized by low nitrogen fixation levels in *A. indica* [72] and in *Vigna unguiculata* [47]. These symbiotic incompatibilities were associated with mutations in the genes encoding the type III secretion system, which is a common and well-described mechanism used not only by bacterial pathogens, but also by bacterial symbionts to suppress plant immune functions [47]. Gully et al. [46] showed that necrotic zones appear in ineffective nodules of *Aeschynomene*, and they are associated with failed rhizobia differentiation.

Some QTL actions described here are close to those previously observed in functional studies performed using RNA interference to downregulate the orthologs of well-known nodulation genes in peanuts. Sinharoy and DasGupta [45] showed that the downregulation of Ca^2+^/calmodulin-dependent protein kinase (*CCaMK*) reduces nitrogenase activity by ~90% and affects rhizobial dissemination during nodule organogenesis. Likewise, the silencing of cytokinin receptor histidine kinase1 (*HK1*) led to a drastic reduction in the nodule number, nodule size, and nitrogenase activity [73]. In our study, we found 17 orthologs of nodulation genes located in the QTL regions, but at this stage, it is difficult to identify which of these genes are responsible for the phenotype(s) due to the large size of the wild introgressed chromosome segments in this CSSL population. A transcriptomic analysis and fine mapping of the major QTLs are underway to identify the underlying gene(s).

### 4.3. Wild Species, Domestication, and BNF

In our study, at all the QTLs, the wild alleles had a depressive effect on the plant growth and nodulation traits, indicating that when the efficient Senegalese strain ISRA400 is used as a symbiotic partner, *A. ipaensis* and *A. duranensis* are not valuable sources of new alleles for increasing BNF. These results suggest either a potential positive impact of domestication and/or the breeding process in the improvement of peanut BNF or a disruption of the symbiotic mechanisms between the wild species and a foreign rhizobia strain as the result of peanut introduction and adaptation in Senegal. Bouznif et al. [74] suggested that cultivated peanuts acquired their current rhizobia by adapting to novel symbionts encountered within the area of introduction. Wild relative species of peanuts originating in South America exhibited a better nitrogen fixation ability when inoculated with native rhizobia strains from Argentina compared to the Senegalese strain *B. vignae* ISRA400 [75]. BNF is possibly the result of a coevolution and/or coadaptation process between symbiotic partners located in the same geographical area. Root nodule bacteria associated with cultivated chickpeas are phylogenetically different from those derived from the nodules of their wild relatives, indicating a coadaptation of rhizobia with legume domestication [76]. In common beans, plants were generally compatible with rhizobia isolated from their domestication centers [77], and specific polymorphisms were found both in bacterial and plant symbiotic genes [78]. However, no evidence for an adaptation to native rhizobia was found in *Medicago lupulina* [79]. In soybeans, a recombinant inbred line (RIL) population derived from a cross between a cultivated and wild germplasm showed that an improved BNF capacity could be associated with cultivated varieties [16]. Recently, the phenotyping of nodulation traits on 310 soybean accessions (10 wild soybeans, 71 landraces, and 229 improved cultivars) and 207 CSSLs allowed for the identification of the SNPs selected during the domestication and located in the regulated regions of the *GmCRP* gene encoding a rhizobial type III effector [60]. We hypothesize that the introgressions of the wild alleles of *A. duranensis* and *A. ipaensis* at the QTL region in the cultivated background of Fleur11 has partially disrupted the symbiotic association with ISRA400 in the absence of coevolution or coadaptation between the Senegalese strain and the wild species. To provide more insights into this assertion, it would be interesting to explore the response of the CSSL population when inoculated with strains coming from South America, where peanuts are native.

## 5. Conclusions

In this study, an interspecific population of 83 chromosome substitution segment lines was used to map genomic regions controlling biological nitrogen fixation traits in peanuts. We found four QTLs that were consistently associated with peanut nodulation. At these QTLs, the wild alleles had a negative effect on biological nitrogen traits. Then, screening these genomic regions allowed for the identification of 17 orthologs of the nodulation genes. However, more studies are needed to identify the underlying gene(s). Our results provide additional insights into the genetic aspects of peanut nodulation and will contribute to a better knowledge of the potential impact of domestication on the enhancement of the nitrogen fixation capacity of peanuts.

## Figures and Tables

**Figure 1 genes-14-00797-f001:**
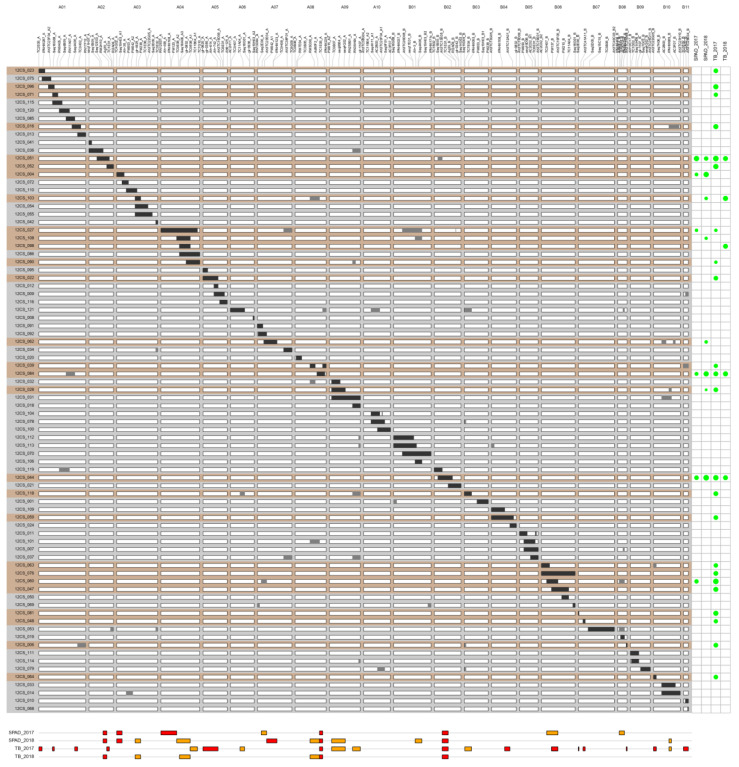
Position of QTLs related to BNF in the genetic map of peanuts. Each column indicates a linkage group named from A01 to A10 (A genome) and B01 to B11 (B genome). Each row indicates a CSSL. The white background represents the genetic background of the recurrent parent Fleur11. The black areas represent the wild target chromosome segments, while the grey areas represent the wild supernumerary chromosome segments. CSSLs showing significant differences compared to Fleur11 are highlighted in orange. The green circles on the left of the figure represent the relative effect on the QTL. Their sizes are proportional to the QTL’s relative effect. SPAD and TB correspond to the leaf chlorophyll content and the total biomass, respectively, evaluated in 2017 and 2018. All the QTLs identified had a negative effect on the measured traits. QTL locations are indicated by rectangles located at the bottom of the figure. Red = QTL with known location, orange = putative QTL.

**Figure 2 genes-14-00797-f002:**
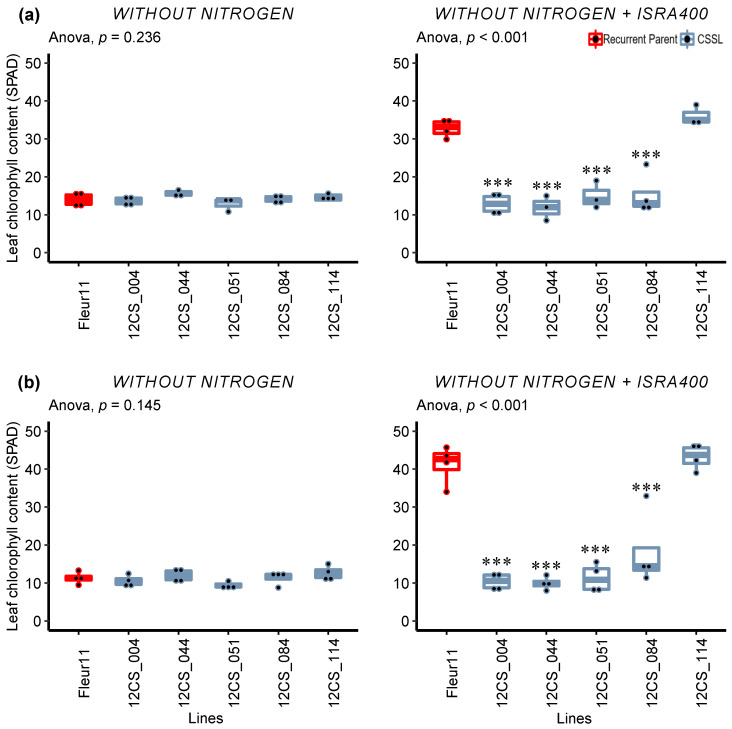
Variation in leaf chlorophyll content 35 days after inoculation under controlled conditions. (**a**,**b**): First and second experiments, respectively. Without nitrogen and without nitrogen + ISRA400 indicate uninoculated plants (negative control) and plants inoculated with *B. vignae* ISRA400, respectively. *** indicate values significant at *p* < 0.001, respectively, using Dunnett’s multiple comparisons test.

**Figure 3 genes-14-00797-f003:**
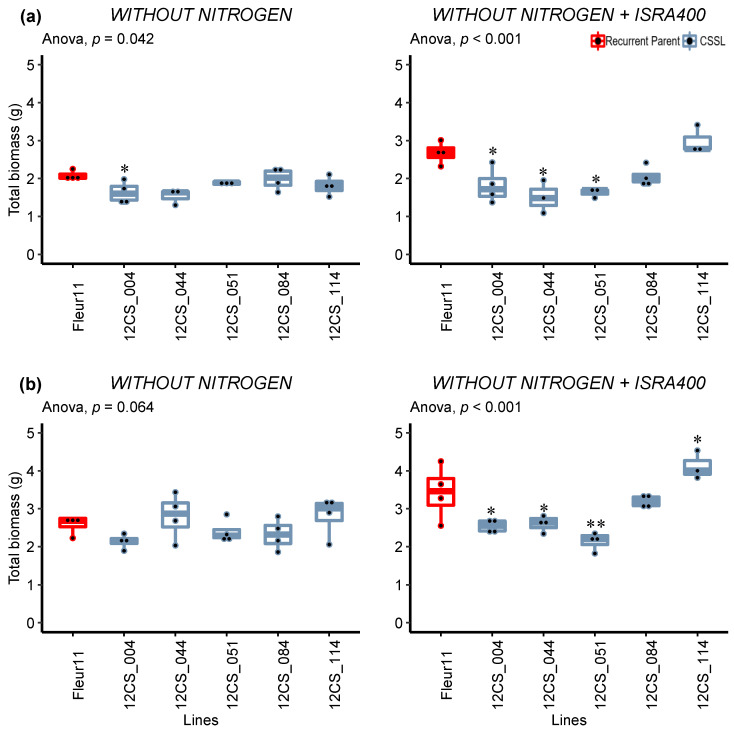
Variation in total biomass 35 days after inoculation under controlled conditions. (**a**,**b**): First and second experiments, respectively. Without nitrogen and without nitrogen + ISRA400 indicate uninoculated plants (negative control) and plants inoculated with *B. vignae* ISRA400, respectively. * and ** indicate values significant at *p* < 0.05 and *p* < 0.01, respectively, using Dunnett’s multiple comparisons test.

**Figure 4 genes-14-00797-f004:**
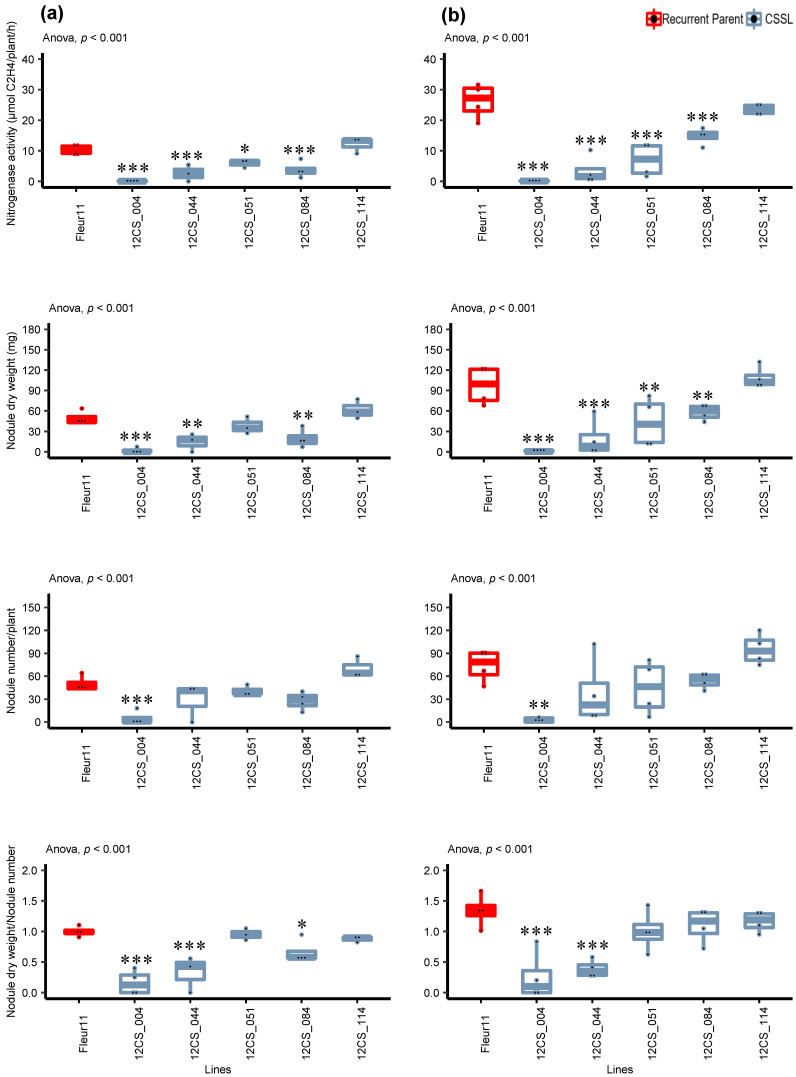
Significant line × trait of nodulation traits 35 days after inoculation between 5 CSSLs and the recurrent parent Fleur11. (**a**,**b**): First and second experiments, respectively. *, **, and *** indicate values significant at *p* < 0.05, *p* < 0.01, and *p* < 0.001, respectively, using Dunnett’s multiple comparisons test.

**Figure 5 genes-14-00797-f005:**
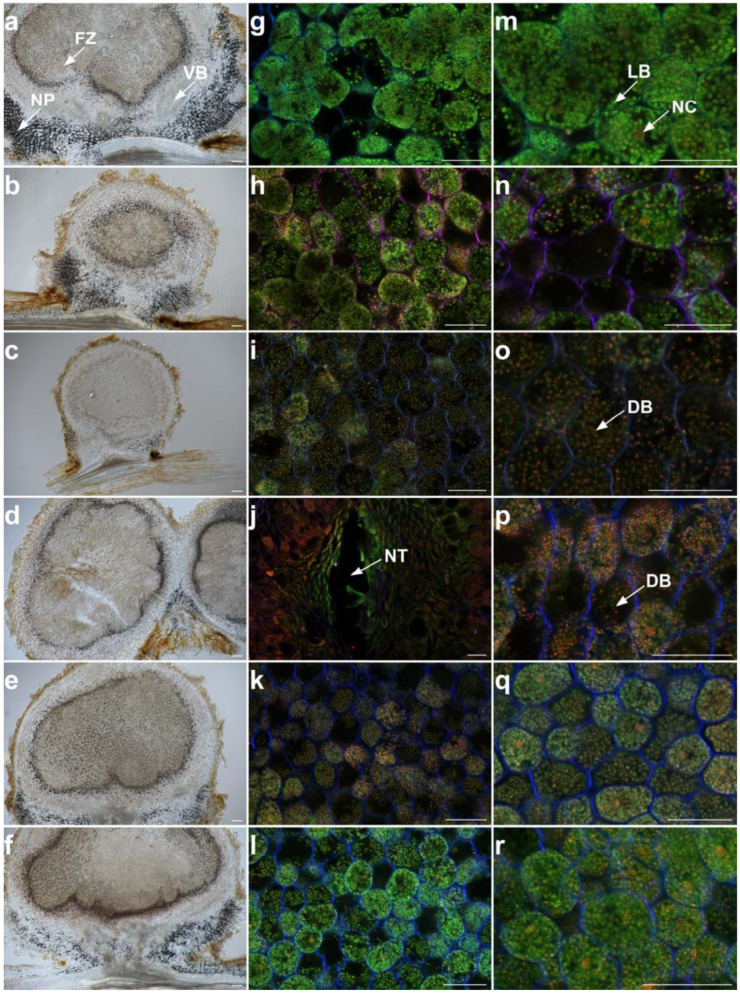
Longitudinal sections of the nodules of 5 CSSLs and the recurrent parent Fleur11 35 days after inoculation with *B. vignae* ISRA400. Differential interference contrast microscopic images (**a**–**f**) showing nodule structure and confocal microscopic images (**g**–**r**) showing live (green) bacteria, dead (red) bacteria, and plant cell walls (blue). Nodule sections of Fleur11 (**a**,**g**,**m**); line 12CS_004 (**b**,**h**,**n**); line 12CS_044 (**c**,**i**,**o**); line 12CS_051 (**d**,**j**,**p**); line 12CS_084 (**e**,**k**,**q**); and line 12CS_114 (**f**,**l**,**r**). FZ: fixation zone; NP: nodule parenchyma or inner cortex; VB: vascular bundles; NC: nucleus; NT: collapsed necrotic tissue; LB: live bacteroids; DB: dead bacteroids. Scale bar = 50 µm.

**Table 1 genes-14-00797-t001:** Descriptive statistics and broad-sense heritability in 2017 and 2018 experiments for traits related to BNF: leaf chlorophyll content and total biomass.

Trait	Treatment	2017				2018			
		Mean ± SD	F	Pr	h^2^	Mean ± SD	F	Pr	h^2^
SPAD	−N	26.2 ± 7.82	1.017	0.456	0	14.8 ± 5.10	1	0.583	0
	+N	43.5 ± 3.79	1.742	0.001 **	0.43	40.6 ± 2.06	2	<0.001 ***	0.50
	−N+ISRA400	41.6 ± 6.42	3.843	<0.001 ***	0.74	39.8 ± 6.48	11.358	<0.001 ***	0.91
TB	−N	2.88 ± 1.26	1.611	0.005 **	0.38	2.69 ± 0.96	2.486	<0.001 ***	0.60
	+N	5.75 ± 1.62	1.310	0.073	0.24	7.98 ± 2.07	1.446	0.023 *	0.30
	−N+ISRA400	4.36 ± 1.63	2.324	<0.001 ***	0.57	3.90 ± 1.47	1.979	<0.001 ***	0.49

SPAD: leaf chlorophyll content 49 days after inoculation; TB: total biomass (g); −N: without nitrogen; +N: with nitrogen; −N+ISRA400: without nitrogen and inoculated with *B. vignae* ISRA400; SD: standard deviation; F: value of genotype effect: Pr: *p*-value associated with the F-value of the fixed-effect genotype; h^2^: broad-sense heritability. *, **, and *** indicate values significant at *p* < 0.05, *p* < 0.01, and *p* < 0.001, respectively.

**Table 2 genes-14-00797-t002:** Summary of the line × trait significant associations for the inoculated treatment in the 2017 and 2018 environments.

Trait	CSSL	LG	2017		2018	
			QTL Effect (%)	Conf. Int (cM)	QTL Effect (%)	Conf. Int (cM)
SPAD	12CS_051	A02	−65.29 ***	43.85–55.35	−43.26 ***	43.85–55.35
	12CS_004	A03	−29.23 ***	0–16.6	−64.77 ***	0–16.6
	12CS_103	A03	ns		−26.76 ***	57.2–74.2
	12CS_027	A04	−24.61 *	0–48.975	ns	
	12CS_108	A04	ns		−22.79 ***	48.975–90.975
	12CS_060	A07	−41.77 ***	12.1–28.525	ns	
	12CS_062	A07	ns		−26.32 ***	28.525–60.0
	12CS_084	A08	−33.79 ***	75.525–85.375	−61.67 ***	75.525–85.375
	12CS_103	A08	ns		−26.76 ***	46.15–75.525
	12CS_028	A09	ns		−15.17 **	7.125–49.375
	12CS_108	B01	ns		−22.79 ***	66.725–87.525
	12CS_044	B02	−47.89 ***	24.55–43.25	−73.49 ***	24.55–43.25
	12CS_060	B06	−41.77 ***	16.275–50.95	ns	
	12CS_060	B08	−41.77 ***	5.25–21.25	ns	
	12CS_028	B10	ns		−15.17 **	48.025–55.525
TB	12CS_023	A01	−48.76 **	0–9.675	ns	
	12CS_096	A01	−61.13 ***	42.325–47.7	ns	
	12CS_071	A01	−41.65 *	42.325–47.7	ns	
	12CS_016	A01	−58.32 **	111.125–119.975	ns	
	12CS_051	A02	−71.51 ***	55.35–63.125	−56.43 *	43.85–55.35
	12CS_052	A02	−62.55 ***	55.35–63.125	ns	
	12CS_103	A03	ns		−60.58 **	57.2–74.2
	12CS_027	A04	−47.38 **	90.975–113.15	ns	
	12CS_090	A04	−45.74 *	90.975–113.15	ns	
	12CS_098	A04	ns		−55.10 *	57.625–90.975
	12CS_022	A05	−47.87 *	0–46.8	ns	
	12CS_118	A06	−52.02 *	29.825–43.925	ns	
	12CS_084	A08	−53.28 **	75.525–85.375	−58.35 *	75.525–85.375
	12CS_103	A08	ns		−60.58 **	46.15–75.525
	12CS_028	A09	−49.2 *	7.125–49.375	ns	
	12CS_118	A09	−52.02 *	72.1–96.1	ns	
	12CS_044	B02	−66.09 ***	24.55–43.25	−55.36 **	24.55–43.25
	12CS_118	B03	−52.02 *	1.725–22.575	ns	
	12CS_059	B04	−49.12 **	40.875–57.225	ns	
	12CS_076	B06	−52.02 **	31.3–50.95	ns	
	12CS_060	B06	−69.63 ***	31.3–50.95	ns	
	12CS_047	B06	−57.13 ***	31.3–50.95	ns	
	12CS_081	B07	−60.27 **	0–2.05	ns	
	12CS_048	B07	−41.04 *	14.5–21.275	ns	
	12CS_006	B08	−50.57 *	26.95–29.8	ns	
	12CS_063	B10	−44.63 **	0–8.5	ns	
	12CS_064	B10	−47.1**	0–8.5	ns	
	12CS_028	B10	−49.2 *	48.025–55.525	ns	
	12CS_039	B11	−39.73 *	0–15.1	ns	

The QTL effect values for SPAD (leaf chlorophyll content) and TB (total biomass) represent the relative difference between the CSSLs and Fleur11 in each environment (2017 or 2018). LG indicates the linkage group, which carries the wild segment responsible for the phenotype. Conf. Int represents the confidence interval of the QTL on the chromosome. *, **, and *** indicate values significant at *p < 0.05*, *p* < 0.01, and *p* < 0.001, respectively, using Dunnett’s multiple comparisons test; ns: not significant.

**Table 3 genes-14-00797-t003:** Relationships between traits related to BNF.

Trait 1	Trait 2	Corr. Coef.
Leaf chlorophyll content	Nitrogenase activity	0.92 ***
	Nodule dry weight	0.89 ***
	Nodule number	0.81 **
	Nodule dry weight and number ratio	0.72 **
Total biomass	Nitrogenase activity	0.84 ***
	Nodule dry weight	0.80 **
	Nodule number	0.74 **
	Nodule dry weight and number ratio	0.62 *

Corr. coef.: Pearson correlation coefficients obtained for 5 CSSLs and the recurrent parent Fleur11 using the average value per genotype. *, **, and *** indicate values significant at *p <* 0.05, *p* < 0.01, and *p* < 0.001.

## Data Availability

The data presented and/or analyzed during the current study are available from the corresponding author upon reasonable request.

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
