# Peer review of "Mapping of QTLs Associated with Biological Nitrogen Fixation Traits in Peanuts (Arachis hypogaea L.) Using an Interspecific Population Derived from the Cross between the Cultivated Species and Its Wild Ancestors"

_genes, 2023, doi:10.3390/genes14040797_

Round 1

Reviewer 1 Report

In this study, genetic mapping was conducted for the identification of genomic regions associated with biological nitrogen fixation in peanut. An interspecific mapping population was used and phenotyping was conducted for various traits. QTL mapping identified several genomic regions associated with biological nitrogen fixation. Further, putative candidate genes were identified in key QTLs which may play important role in nitrogen fixation. The research provided important information regarding the QTLs and candidate genes associated with biological nitrogen fixation in peanut. Following points may be considered.

1.      Information regarding the genotyping and marker used in this study may be included in the text and as supplementary tables.

2.      Brief information regarding candidate genes may be added.

3. Recently published articles related to the work may be cited and briefly discussed.

Reviewer 2 Report

1)        Line 53: ‘Legumes’à’legumes’

2)        Line 58:’allowed’à’allow’

3)        Lines 484-488: The descriptions ‘We screened the sequences of genomic …’ in the discussion section should be moved to the results section. The information supplied by the Supplementary Table S4 should also be mentioned in the results section.

4)        According to your findings, the line 12CS_004 only possessed one BNF QTL on B03 while the line 12CS_051 possessed three QTLs on A02, A08 and B02. Since these QTLs were additive, why did the line 12CS_004 displayed more significant phenotypic decrease than the line 12CS_051?

5)        Better provide the phenotypes of the male parent AiAd (A. ipaensis × A. duranensis) of BNF traits. Why did you chose these two parental lines to construct the CSSL population ?

Round 2

Reviewer 1 Report

The revised manuscript provides important information regarding the QTLs and candidate genes associated with biological nitrogen fixation traits in peanut. All the suggested corrections have been made in this manuscript. The manuscript is high scientific quality and may be accepted for its publication in this journal.